# PHI-AGENT: A DYNAMIC IN-GRAPH REASONING AGENT INSPIRED BY THE PREFRONTAL–HIPPOCAMPAL INTERACTION

## ABSTRACT

Graph-based Retrieval-Augmented Generation has demonstrated strong performance in multi-hop reasoning and cross-document evidence integration. However, existing methods typically rely on static, one-shot retrieval strategies, lacking the ability to assess evidence sufficiency or dynamically construct context—thereby limiting their effectiveness in complex reasoning tasks. To address these limitations, we propose Phi-agent, a brain-inspired iterative in-graph retrieval agent. Motivated by the hippocampus–prefrontal interaction in cognitive neuroscience, Phi-agent operates in a "retrieve–reason–re-retrieve" loop, enabling proactive refinement of contextual evidence through iterative questioning and reasoning. We further introduce a joint reward mechanism that simultaneously optimizes both reasoning quality and retrieval trajectory. To support reinforcement learning, we curate a high-quality dataset of 7,405 annotated samples and post-train Qwen3-1.7B using the Group Relative Policy Optimization (GRPO) algorithm. Experiments on HotpotQA, MuSiQue, and 2WikiQA show that Phi-agent significantly outperforms existing GraphRAG baselines, achieving state-of-the-art performance. Ablation studies confirm the essential role of the iterative in-graph retrieval loop and joint reward design in enabling these improvements.

## 1 INTRODUCTION

In recent years, with the rapid development of large language models, Retrieval-Augmented Generation (RAG) has emerged as a key approach to mitigating hallucinations and enhancing factual accuracy and interpretability (Arslan et al., 2024; Singh et al., 2025; Yang et al., 2025). Among the various RAG variants, graph-based Retrieval-Augmented Generation (GraphRAG) has garnered significant attention (Sarthi et al., 2024; Sun et al., 2023; Han et al., 2025). By representing entities, relationships, and document passages as knowledge graphs, GraphRAG enables more efficient organization and indexing of knowledge in complex multi-hop reasoning tasks across documents. At the same time, RAG systems are evolving toward context engineering—focusing not only on retrieval itself but also on dynamically selecting and constructing appropriate input contexts in a task-driven manner.

Although GraphRAG exhibits unique advantages in knowledge organization and retrieval, most existing approaches adopt a "one-shot, static" retrieval and context construction paradigm—executing a single retrieval upon receiving a query and directly concatenating the top-K retrieved passages as input to the model. While this approach is efficient for simple tasks, it often underperforms in complex reasoning scenarios, where key pieces of evidence may be missing, leading to broken chains of reasoning. Fundamentally, these methods lack sufficient self-assessment capabilities: they cannot determine whether the current evidence is sufficient to answer the question, nor can they proactively generate new retrieval goals when evidence is lacking. This limitation in context engineering restricts the applicability of GraphRAG.

To address these limitations, this paper proposes Phi-agent, a graph-iterative retrieval agent equipped with dynamic reflection and iterative retrieval capabilities. Inspired by the hippocampus–prefrontal cortex interaction mechanism in cognitive neuroscience, as shown in Fig.1, the agent mimics the process by which the hippocampus retrieves and activates memory, linking new input with existing

memory fragments to support the prefrontal cortex, while the prefrontal cortex performs high-level decision-making, planning, and logical reasoning—evaluating retrieved information and determining the next retrieval steps. This synergy enables the human brain to dynamically access memory and reason during complex cognitive tasks. Similarly, Phi-agent implements a "retrieve–reason–re-retrieve–re-reason" loop: each round begins with hippocampus-like rapid retrieval from the knowledge graph to recall relevant subgraphs and textual information, followed by prefrontal-style reasoning to assess whether the retrieved content is sufficient and whether further retrieval is necessary. Through this dynamic reflection and iterative retrieval, the agent constructs more precise and complete contexts.

Furthermore, the paper introduces a joint reward mechanism guided by both reasoning content and retrieval trajectory. A high-quality training dataset comprising 7,405 samples was constructed through a carefully designed data generation and annotation pipeline. Based on this dataset, the model was trained using the GRPO algorithm (Shao et al., 2024). Experimental results show that the joint reward mechanism effectively regulates the retrieval trajectory—deciding whether to continue retrieving or proceed to answering—thereby optimizing the retrieve–reason strategy loop and significantly enhancing robustness and generalization in complex tasks. On three multi-hop QA and complex reasoning benchmarks—HotpotQA (Yang et al., 2018), (Trivedi et al., 2022), and 2WikiQA (Ho et al., 2020)—Phi-agent demonstrates significantly higher accuracy than existing GraphRAG methods.

In summary, the main contributions of this paper are as follows: 1. Proposing a brain-inspired graph-iterative retrieval agent, Phi-agent, that enables dynamic context retrieval and construction; 2. Introducing a joint reinforcement learning reward mechanism guided by reasoning content and retrieval trajectory, along with contributing a high-quality dataset; 3. Demonstrating through experiments that Phi-agent significantly improves performance across multiple datasets, achieving state-of-the-art results.

## 2 RELATED WORK

### 2.1 PREFRONTAL–HIPPOCAMPAL INTERACTION

A substantial body of neuroscientific research has demonstrated functional coupling between the hippocampus and the prefrontal cortex in the service of memory and planning. First, cross-regional oscillatory interactions—such as theta–gamma coupling between the hippocampus and medial prefrontal cortex—are thought to coordinate the maintenance of working memory and the suppression of interference, thereby providing a mechanistic basis for information selection and updating. Such evidence has been consistently observed in both animal electrophysiology and human neuroimaging studies(Daume et al., 2024; Benchenane et al., 2010; Jin & Maren, 2015). Second, replay or sequential reactivation(Shin & Jadhav, 2016) has been identified as a key process linking stored memory traces with ongoing goals. Recent theoretical models propose that prefrontal task dynamics can trigger hippocampal replay, which in turn shapes prefrontal planning processes, thereby supporting goal-directed exploration and decision-making. More recent work has situated replay within a normative framework for near-optimal exploration. Third, beyond theta oscillations, coordination during sharp-wave ripples (SWRs) between the PFC and hippocampus has also been shown to contribute to memory-guided behavior, providing circuit-level evidence for iterative retrieval–evaluation–updating loopsPreston & Eichenbaum (2013); den Bakker et al. (2023); Patai & Spiers (2021).

From a computational perspective, these findings converge on two insights: (A) the hippocampus serves as a substrate for rapid associative retrieval and episodic fragment activation; and (B) the prefrontal cortex supports task-level evaluation, decision-making, and subgoal generation. These insights provide a biologically grounded analogy for the design of retrieval–reasoning systems in artificial intelligence.

### 2.2 GRAPH-BASED RETRIEVAL-AUGMENTED GENERATION

Compared with vector-database-based RAG, GraphRAG explicitly models entities, relations, and inter-fragment connections, emphasizing multi-hop retrieval across documents and structural interpretability. The GraphRAG(Edge et al., 2024) pipeline introduced by Microsoft integrates document

extraction, graph construction, and community- or subgraph-level summarization, enhancing contextual organization and answer quality through a community aggregation → divide-and-conquer → synthesis process. Subsequently, multiple studies have systematized GraphRAG research, organizing its retrieval primitives, indexing strategies, alignment methods, and evaluation metrics, thereby providing a comparative and terminological framework.

Representative approaches include HippoRAG(Jimenez Gutierrez et al., 2024), inspired by the hippocampal indexing theory, which synergizes large language models, knowledge graphs, and personalized PageRank (PPR) to enhance long-range integration via fast indexing plus graph-guided retrieval, marking a milestone in connecting neuro-inspired mechanisms with GraphRAG. LightRAG(Guo et al., 2024) advocates for a simple yet efficient two-level retrieval scheme—combining fine-grained fragments with higher-level structural summaries—achieving engineering efficiency while incorporating graph-structural benefits. G-RetrieverHe et al. (2024) formalizes RAG over graphs as a reward-regularized Steiner subtree retrieval and generation problem, yielding scalable solutions for open-domain text-graph question answering. KGPWang et al. (2024) leverages knowledge graph construction and traversal to seamlessly integrate global structural constraints with local semantic navigation, thereby significantly enhancing prompt design and performance in multi-document question answering. DALK(Li et al., 2024), targeting biomedical and other long-tail knowledge domains, proposes dynamic co-enhancement between LLMs and domain knowledge graphs, leveraging temporal graph evolution and self-aware retrieval to improve specialized QA performance.

Despite these advances in graph modeling and graph-guided retrieval, most existing approaches rely on static, one-shot retrieval followed by top-$K$ concatenation. Such designs often lack mechanisms for self-assessment of evidence sufficiency and iterative query refinement, leaving them vulnerable to reasoning failures when critical evidence is absent or when retrieval paths are incomplete.

## 3 METHOD

### 3.1 PREFRONTAL–HIPPOCAMPAL INTERACTION INSPIRED AGENT

To address the limitations of the "one-shot, static" retrieval strategy in traditional GraphRAG systems for multi-hop question answering and complex reasoning tasks, we design Phi-agent—a brain-inspired agent with reflective mechanisms that supports iterative retrieval over knowledge graphs. This design draws inspiration from the "prefrontal–hippocampal interaction theory" in cognitive neuroscience. The agent simulates how the human brain integrates information and makes decisions through the synergy of memory activation and logical evaluation when faced with complex cognitive tasks, enabling multi-round retrieval, evaluation, and dynamic context construction.

In the agent's workflow, a graph retrieval algorithm based on Personalized PageRank (PPR) simulates the function of the hippocampus to recall relevant information $C_t$ from the knowledge graphJimenez Gutierrez et al. (2024). Given the current query goal, this process rapidly activates related entity nodes and semantic paths from the knowledge graph, producing a set of semantically related graph information:

$$C_t = \{c_1, c_2, \ldots, c_k\}$$

These items form part of the agent's "working memory" for the $t$-th round of retrieval or reasoning and are passed to the language model for further analysis.

The proposed Phi-agent corresponds to the prefrontal cortex (PFC), and its core task is to assess whether the information in working memory is sufficient to answer the original query $q_0$. If the assessment concludes that the information is "sufficient," the system outputs the answer $\hat{a}$ directly; otherwise, the model identifies the missing information and generates a supplemental sub-query $q_{t+1}$ to guide the next round of graph retrieval.

The complete process of Phi-agent is as follows: in each round $t$, the system uses the sub-query $q_t$ to retrieve relevant information $C_t$ from the knowledge graph using the PPR algorithm.

To enhance cross-round semantic coherence and reduce redundancy, the system applies a semantic deduplication strategy each round, comparing the current context $C_t$ with previously retrieved contexts $C_{<t}$ and retaining only novel, informative fragments. The cumulative working memory is

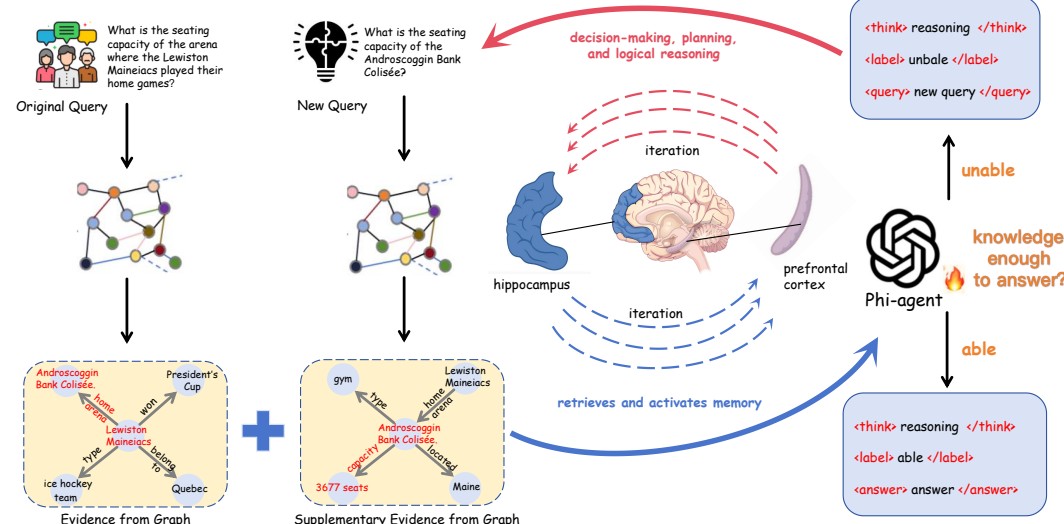

Figure 1: Phi-agent operates in a "retrieve–reason–re-retrieve" loop, enabling proactive refinement of contextual evidence through iterative questioning and reasoning

then:

$$C_{\leq t} = C_{<t} \cup \mathrm{Dedup}(C_t, C_{<t}) \tag{1}$$

Once the language model receives the graph information, it performs reflective reasoning based on a structured prompt format. The model must assess whether the accumulated context is sufficient to complete the original task $q_0$ and output results in a structured format. As shown in Fig.1. If sufficient, the output contains the tag ¡label¿ able ¡/label¿ and the final answer.

If the information is insufficient, it outputs ¡label¿ unable ¡/label¿ and a new sub-query ¡query¿ to trigger the next retrieval iteration:

By simulating hippocampal memory activation and prefrontal planning, Phi-agent enables dynamic exploration of multi-hop paths in the knowledge graph and self-driven supplementary querying. This mechanism improves the relevance and coverage of retrieval contexts and significantly enhances reasoning depth and logical completeness in complex, multi-turn reasoning tasks.

All prompts related to this paper are detailed in the appendix.

## 3.2 RL GUIDED BY RETRIEVAL TRAJECTORY AND REASONING CONTENT

### 3.2.1 DATASET CONSTRUCTION

To train an agent capable of dynamic retrieval and reasoning over knowledge graphs, we construct a reinforcement learning dataset based on HotpotQA. The dataset construction process consists of three stages: initial sample generation, llm-based label annotation, and graph structure completion.

First, we use the HippoRAG algorithm to extract entities and build relationships from the knowledge base provided by HotpotQA, resulting in a unified knowledge graph. Next, HippoRAG is used as the initial retriever to perform graph-guided retrieval on the original query $q_0$, obtaining a set of relevant passages as initial context $C_0$ along with the gold answer. These form the initial training triples: {*original query*, *retrieved context*, *gold answer*}.

To determine whether the retrieved context supports answering the question, we introduce a multi-model voting mechanism for label annotation. Specifically, we use three large language models(DeepSeek-V3-0324, GPT-4o, and Gemini-2.5-pro) to assess each sample. If the majority determine that the context supports correct reasoning to the answer, the sample is labeled as *label = able*, and the gold answer is retained. Otherwise, if the context is deemed insufficient, the sample is labeled as *label = unable*, and DeepSeek-V3-0324 is further used to generate a supplemental sub-query $q_1$ to guide the next retrieval step.

Ultimately, we obtain two types of samples labeled by the voting mechanism. Sufficient information samples: {*original query, retrieved context, label=able, gold answer*} and insufficient information samples: {*original query, retrieved context, label=unable, next query*}

### 3.2.2 REWARD FUNCTION DESIGN

To improve the agent's performance in the ""retrieve–reason–re-retrieve" loop, we design a joint reward mechanism to guide and provide feedback on Phi-agent's dynamic retrieval behavior from three dimensions: reasoning content, retrieval trajectory, and output formatting.

**Retrieval Trajectory Reward** In each iteration, the agent must decide whether the current context is sufficient to answer the question and whether to initiate a new query—i.e., classify between *able* and *unable*. We use reference labels obtained via majority vote from GPT-4o, Gemini-2.5-pro, and DeepSeek-V3-0324. If the agent's decision matches the reference, it receives a reward $r_{\text{label}}$; otherwise, no reward is given. This helps optimize the retrieval trajectory.

**Content Reward** For samples labeled *able*, the final answer $\hat{a}$ generated by the agent is compared with the gold answer using the LLaMA3.3-70B model to determine correctness. If correct, a reward of $r_{\text{ans}} = 1$ is assigned; otherwise, $r_{\text{ans}} = 0$. For samples labeled *unable*, we compare the generated sub-query $\hat{q}_{t+1}$ with a gold supplemental query $q_{t+1}^{\text{gold}}$ from the dataset using LLaMA3.3-70B. If they are semantically similar, a reward of 1 is given; otherwise, 0. The reasonableness of the sub-query also contributes to optimizing retrieval trajectory. To ensure feedback efficiency, the LLaMA3.3-70B model is restricted to output at most one token.

**Formatting Reward** To ensure structured and parsable reasoning outputs, we enforce output formatting rules that require the use of standardized tags such as *<think>* , *<label>* , and either *<answer>* or *<query>* . If the output format is valid, an additional reward $r_{\text{fmt}} = 1$ is granted; otherwise, no formatting reward is given.

**Joint Reward** The final joint reward function is defined as:

$$R = \alpha \cdot r_{\text{ans}} + \beta \cdot r_{\text{fmt}} + \gamma \cdot r_{\text{label}} \tag{2}$$

where $\alpha$, $\beta$, and $\gamma$ are weight coefficients balancing the contributions of different sub-goals to the overall training objective.

### 3.2.3 OPTIMIZATION OBJECTIVE

Smaller LLM could offer advantages in practical retrieval tasks, including faster inference and easier deployment on edge devices. Therefore, we adopt the GRPO algorithm to train the Qwen3-1.7B model. GRPO compares the relative advantages among multiple candidate actions under the same state, avoiding bias from value function estimation and making it suitable for complex strategy optimization in natural language generation.

Let the agent policy be $\pi_\theta(a \mid s)$, where $s$ denotes the current state, including the query $q_0$ and retrieved information $C_{\leq t}$, and $a$ is the complete structured output sequence comprising the tags *<think>* , *<label>* , and *<answer>* or *<query>* . The training objective is to maximize the expected reward:

$$J(\theta) = \mathbb{E}_{a \sim \pi_\theta} [R(s, a)] \tag{3}$$

where $R(s, a)$ is the joint reward function. For each state $s_j$, we sample $K$ candidate responses $\{a_{j1}, \ldots, a_{jK}\}$ and compute their rewards $R_{jk}$ and the average reward $\bar{R}_j$. The advantage function is then:

$$A_{jk} = R_{jk} - \bar{R}_j, \quad \text{where } \bar{R}_j = \frac{1}{K} \sum_{k=1}^{K} R_{jk} \tag{4}$$

The overall loss function is:

$$\mathcal{L}_{\text{GRPO}} = -\sum_{j,k} A_{jk} \cdot \log \frac{\pi_\theta(a_{jk} \mid s_j)}{\pi_{\theta_{\text{old}}}(a_{jk} \mid s_j)} + \delta \cdot \text{KL} \left[ \pi_\theta(\cdot \mid s_j) \| \pi_{\theta_{\text{old}}}(\cdot \mid s_j) \right] \tag{5}$$

where $\pi_{\theta_{\text{old}}}$ is the frozen reference policy, and $\beta$ is the KL regularization coefficient to constrain policy drift and ensure training stability.

# 4 EXPERIMENTS

## 4.1 SETUP

**Dataset**   We evaluate our proposed method on three widely used multi-hop question answering benchmarks: HotpotQA, MuSiQue, and 2WikiMultiHopQA. Following the experimental protocol introduced in HippoRAG, we construct evaluation subsets by uniformly sampling 1,000 queries from each dataset. For each subset, a corresponding passage collection is assembled, containing both supporting and distractor evidence from the original corpus. The resulting passage corpora contain 9,811 entries for HotpotQA, 11,656 for MuSiQue, and 6,119 for 2WikiMultiHopQA.

In addition, we construct a reinforcement learning dataset based on the full HotpotQA corpus, following the data generation procedure described in Section. The resulting dataset consists of 7,405 instances, with 6,405 used for training and 1,000 for testing. The test set shares the same *original query* instances as the HotpotQA evaluation subset.

Each data instance includes four fields: *original query*, *retrieved context*, *label* (either *able* or *unable*), and the corresponding *gold answer* or *next query*. Within the training set, 3,496 instances are labeled as *able* with fields *original query, retrieved context, label=able, gold answer*, while 2,909 instances are labeled as *unable* with fields *original query, retrieved context, label=unable, next query*. The test set consists of 550 able-labeled and 450 unable-labeled instances, respectively.

**Baseline**   We include five representative retrieval-augmented generation (RAG) methods as baselines in our experiments: HippoRAG, KGP, DALK, LightRAG, and G-Retriever, all originally proposed within the GraphRAG framework. These methods reflect a range of structured and graph-based retrieval strategies for multi-hop question answering.

To ensure a fair and consistent comparison, we reproduce all baselines under a unified experimental setup and evaluate them using the same metrics. We employ Qwen-1.7B, Qwen-4B, Qwen-8B, Qwen-14B and Qwen3-32B as the backbone language models, along with the all-MiniLM-L6-v2 embedding model for retrieval. All baseline implementations are built on top of the DIGIMON open-source libraryZhou et al. (2025) to ensure reproducibility and implementation consistency.

**Evaluation**   We evaluate our method along the following key dimensions:

- *Accuracy(ACC)*: This metric measures the proportion of correctly answered questions over the entire evaluation set, regardless of whether the agent determines the question to be answerable.

- *Reasoning Accuracy(R-ACC)*: For our proposed Phi-agent, we further assess its performance on the subset of samples where the agent predicts the query to be answerable. Within this subset, we calculate the proportion of correct answers, reflecting the agent's conditional accuracy when it believes it can answer—thus providing a more faithful evaluation of its multi-hop reasoning capability.

- *Response Token Cost*: The average number of tokens generated per sample throughout the entire reasoning process until a final answer is produced.

- *Average Iteration Number*: The average number of retrieval–reasoning iterations executed by the agent before arriving at a final decision.

To evaluate both ACC and R-ACC, we employ multiple models for automatic answer judgment, including *GPT-4o*, *DeepSeek-V3-0324*, and *Gemini-2.5-Pro*. In our experiments, we observe that these state-of-the-art models exhibit high consistency when determining the correctness of predicted answers given a query and its corresponding gold answer. Therefore, we report the average accuracy obtained from the three models as the final evaluation metric.

## 4.2 PERFORMANCE COMPARISON

Our method is built upon the HippoRAG framework and further incorporates the Phi-agent, combined with reinforcement learning guided by retrieval trajectories and reasoning content. With this design, we achieve significant performance improvements across three multi-hop question answering benchmarks: HotpotQA, 2Wiki, and MuSiQue. Specifically, our method achieves state-of-the-art performance in both top-10 and top-20 accuracy. As shown in Table 1, our approach improves the top-10 accuracy by more than 20 percentage points over HippoRAG on HotpotQA and 2Wiki, and by over 13 percentage points on MuSiQue. These consistent gains strongly demonstrate the effectiveness of our retrieval-aware reasoning framework in tackling complex multi-hop questions.

The superior performance stems from: (1) the dynamic iterative retrieval strategy of Phi-agent, which enables more comprehensive and targeted evidence acquisition; (2) the reinforcement learning guided by retrieval trajectory and reasoning content, which effectively steers the model toward higher-quality iterative reasoning paths. Notably, as illustrated in Fig. 2, our method exhibits a highly practical trend: using only a small-scale model (HippoRAG-$\phi$ with 4B parameters) already surpasses the performance of the original HippoRAG with a 32B model. This efficiency advantage highlights the deployment potential and practicality of our method in real-world scenarios.

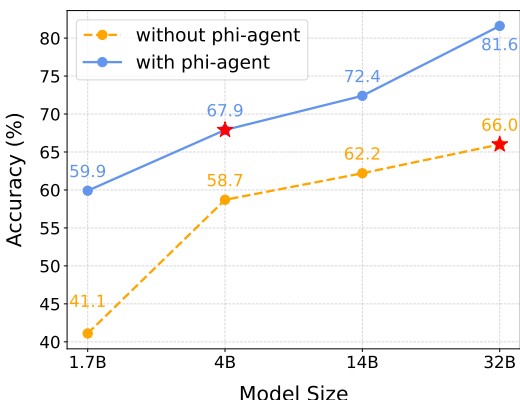

Figure 2: Accuracy trend with respect to model size. HippoRAG-$\phi$ (4B) outperforms HippoRAG (32B).

## 4.3 ABLATION STUDIES

**Effectiveness of Phi-agent** The Phi-agent is designed as a highly modular plug-and-play component, compatible with a wide range of existing retrieval-augmented generation (RAG) methods. Its core objective is to enhance the reasoning capability and retrieval robustness of the original GraphRAG framework by introducing an iterative retrieval mechanism, all without requiring structural modifications to the underlying models.

To comprehensively evaluate its generalizability and transferability, we integrate Phi-agent into several representative baseline models, resulting in five composite systems: Dalk-$\phi$, LightRAG-$\phi$, G-Retriever-$\phi$, HippoRAG-$\phi$, and KGP-$\phi$. These baselines cover a diverse set of graph structures, retrieval path designs, and control logics, enabling us to objectively assess the adaptability of Phi-agent across different GraphRAG scenarios.

As shown in Table2, Experimental results under the retrieval parameter of top-5 and a maximum of 5 iterations demonstrate that the introduction of Phi-agent consistently improves answer accuracy across all composite models and maintains stable performance gains on multiple datasets. Notably, the improvement is especially prominent in small and medium-sized models (e.g., Qwen-1.7B and Qwen-4B), where Phi-agent even enables some configurations to outperform their larger counterparts without Phi-agent (e.g., Qwen-32B). These findings highlight the strong structural compatibility of Phi-agent and its ability to significantly enhance the reasoning depth and decision quality of the models it augments.

**Effectiveness of Joint Reward** To further verify whether our proposed joint reward mechanism can effectively optimize the "retrieve–reason–re-retrieve" process, we conduct ablation studies using LightRAG-$\phi$ as the base framework. $\alpha$, $\beta$,$\gamma$ are set to 1. The agent is allowed to freely perform multi-round retrieval and reasoning without any predefined iteration limits nder the retrieval parameter of top-10. We then analyze the number of questions correctly answered within each of the first eight iterations.

On the untrained Qwen3-1.7B model, as shown in Fig.4 .we observe that 554 samples can be correctly answered in the first round, with the number of correctly answered questions in the 2nd to

Table 1: Comparison of methods on HotpotQA, Wiki, and Musique datasets.

| Method | HotpotQA | | Wiki | | Musique | | Avg | |
|---|---|---|---|---|---|---|---|---|
| | top-10 | top-20 | top-10 | top-20 | top-10 | top-20 | top-10 | top-20 |
| dalk | 20.5 | 20.6 | 22.7 | 24.3 | 5.4 | 7.1 | 16.2 | 17.3 |
| LightRAG | 44.7 | 51.4 | 23.5 | 29.7 | 17.3 | 18.6 | 28.5 | 33.2 |
| G-Retriever | 11.5 | 10.2 | 6.5 | 6.3 | 3.0 | 3.8 | 7.0 | 6.8 |
| KGP | 34.8 | 41.7 | 15.5 | 20.3 | 12.8 | 13.7 | 21.0 | 25.2 |
| HippoRAG | 39.6 | 41.9 | 38.4 | 40.1 | 13.1 | 13.0 | 30.4 | 31.6 |
| **Ours** | **61.7** | **63.3** | **60.7** | **23.6** | **23.6** | **25.7** | **48.7** | **50.57** |

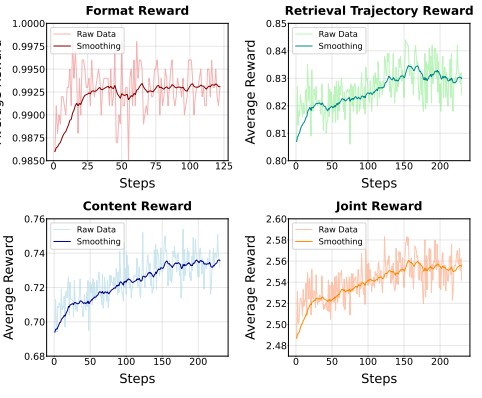

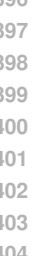
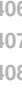

Figure 3: Reward curves over training steps.

Figure 4: R-ACC curves over retrieval iterations.

8th rounds being 174, 79, 40, 28, 20, 14, and 14, respectively. However, as the number of iterations increases, the R-ACC drops sharply. This decline occurs because the remaining questions tend to be more challenging, requiring harder-to-retrieve evidence and longer reasoning chains.

After training Qwen3-1.7B on our carefully constructed reinforcement learning dataset, the resulting model—Qwen3-1.7B-$\alpha$—exhibits significantly improved performance across all iterations. The reward trend is shown in Fig.3. Experimental results under the retrieval parameter of top-10 and a maximum of 5 iterations demonstrate that The R-ACC increases substantially and remains more stable, effectively mitigating the degradation observed in the untrained model during later iterations.

This improvement can be attributed to our proposed joint reward mechanism. On one hand, the retrieval trajectory reward encourages the model to learn more optimal retrieval paths and enhances its ability to judge whether the retrieved information is sufficient. On the other hand, the content reward guides the model in generating more logical and complete answers when it believes it can respond, and in proposing more appropriate follow-up queries when it decides it cannot answer, thus optimizing the overall retrieval path. Additionally, as shown in Table.3. we observe increases in both the response token cost and the average number of iterations after RL training, indicating that the model's reasoning depth and decision complexity have also been enhanced. Ultimately, the overall answer accuracy of LightRAG-$\phi$ improves from 61.3% to 66.5%, validating the effectiveness of our reward modeling strategy in multi-hop question answering scenarios.

## 5 CONCLUSION

This paper introduces a neuro-inspired framework for graph-based retrieval-augmented generation, motivated by the functional interplay between the prefrontal cortex and the hippocampus. Drawing on the division of labor in neural circuits—where the hippocampus supports rapid associative re-

Table 2: Ablation of phi-agents on HotpotQA, 2Wiki, and Musique. "$\phi$" indicates the use of phi-agent.

| Method | HotpotQA | | | 2Wiki | | | Musique | | |
|---|---|---|---|---|---|---|---|---|---|
| | 1.7B | 4B | 32B | 1.7B | 4B | 32B | 1.7B | 4B | 32B |
| Dalk | 21.6 | 32.1 | 45.8 | 23.3 | 26.0 | 27.4 | 6.3 | 11.7 | 21.1 |
| Dalk-$\phi$ | 23.9 | 35.6 | 52.0 | 27.2 | 35.9 | 49.0 | 8.5 | 19.2 | 25.3 |
| LightRAG | 38.8 | 46.7 | 53.4 | 18.9 | 19.3 | 21.3 | 15.4 | 21.5 | 28.9 |
| LightRAG-$\phi$ | 59.5 | 76.0 | 86.3 | 53.3 | 68.0 | 77.8 | 27.8 | 45.4 | 58.0 |
| G-Retriever | 12.0 | 19.9 | 38.0 | 5.6 | 8.6 | 21.7 | 3.0 | 7.2 | 17.3 |
| G-Retriever-$\phi$ | 17.1 | 29.1 | 42.2 | 20.3 | 27.0 | 31.8 | 5.5 | 10.0 | 20.4 |
| HippoRAG | 41.1 | 58.7 | 66.0 | 40.9 | 55.9 | 60.9 | 13.1 | 23.2 | 31.6 |
| HippoRAG-$\phi$ | 59.9 | 67.9 | 81.6 | 57.5 | 74.7 | 80.8 | 23.5 | 41.4 | 52.4 |
| KGP | 31.4 | 39.7 | 51.2 | 17.9 | 23.3 | 31.6 | 9.5 | 13.2 | 24.6 |
| KGP-$\phi$ | 38.5 | 46.6 | 59.1 | 24.4 | 30.0 | 35.4 | 15.5 | 21.9 | 32.9 |

Table 3: Response token cost and average iteration numbers on HotpotQA, 2Wiki, and Musique.

| Dataset | Method | Response Token Cost | | Avg Iteration | |
|---|---|---|---|---|---|
| | | 1.7B | 1.7B-$\alpha$ | 1.7B | 1.7B-$\alpha$ |
| HotpotQA | Dalk-$\phi$ | 10563 | 11953 | 1.81 | 2.13 |
| | HippoRAG-$\phi$ | 5655 | 8501 | 1.72 | 2.38 |
| | KGP-$\phi$ | 6693 | 8249 | 1.61 | 2.21 |
| | LightRAG-$\phi$ | 11065 | 12663 | 2.08 | 2.53 |
| 2Wiki | Dalk-$\phi$ | 13298 | 13944 | 2.12 | 2.65 |
| | HippoRAG-$\phi$ | 4240 | 8968 | 1.64 | 2.66 |
| | KGP-$\phi$ | 7961 | 9357 | 1.68 | 2.39 |
| | LightRAG-$\phi$ | 17613 | 23042 | 1.58 | 2.23 |
| Musique | Dalk-$\phi$ | 11165 | 13933 | 1.97 | 2.57 |
| | HippoRAG-$\phi$ | 7608 | 8923 | 1.85 | 2.19 |
| | KGP-$\phi$ | 8723 | 10653 | 1.91 | 2.32 |
| | LightRAG-$\phi$ | 16805 | 22368 | 1.48 | 2.14 |

trieval and the prefrontal cortex is responsible for evaluation, planning, and subgoal generation—we design an iterative "retrieve–reason–re-retrieve" paradigm within graphs. In addition, we incorporate joint reward signals to optimize both answer accuracy and retrieval trajectory, thereby achieving greater robustness in multi-hop reasoning over knowledge graphs. Experimental results on benchmark datasets such as HotpotQA, MuSiQue, and 2WikiMultiHopQA demonstrate that our approach consistently outperforms strong GraphRAG baselines. By integrating biologically inspired principles with graph structures, our framework highlights a promising direction for retrieval-augmented systems, not only improving accuracy but also enabling smaller models to surpass larger ones, thereby lowering deployment requirements. Looking ahead, we plan to further investigate context-construction strategies in iterative retrieval paradigms and explore parameterizing knowledge to further reduce token consumption and reasoning latency.

## 6    ETHICS STATEMENT

This research complies with the ICLR Code of Ethics. No human subjects or animal experiments were involved in this study. All datasets were obtained in strict accordance with relevant usage guidelines, ensuring no violation of privacy regulations. We carefully avoided any potential biases or discriminatory outcomes during the research process. No personally identifiable information was used, and no experiments were conducted that could raise privacy or security concerns. We are committed to maintaining transparency and academic integrity throughout the research process.

## 7    REPRODUCIBILITY STATEMENT

We have made every effort to ensure that the results presented in this paper are reproducible. All code and datasets have been made publicly available in an anonymous repository: `https://anonymous.4open.science/r/Phi-agent/README.md`, enabling others to replicate and verify our findings. The experimental setup, including training steps, model configurations, and hardware details, is described in detail in the paper. Furthermore, the three public datasets used in this work are all openly accessible, ensuring consistency and reproducibility of evaluation results.

## 8    LLM USAGE STATEMENT

Large Language Models (LLMs) were employed to assist in the writing and polishing of this manuscript. Specifically, we used an LLM to help improve language expression, enhance readability, and ensure clarity across different sections of the paper. The model assisted with tasks such as sentence rephrasing, grammar checking, and improving the overall flow of the text.

It is important to emphasize that the LLM was not involved in the research ideation, methodological design, or experimental implementation. All research concepts, ideas, and analyses were independently developed and conducted by the authors. The contributions of the LLM were strictly limited to improving the linguistic quality of the paper, with no involvement in scientific content or data analysis.

The authors take full responsibility for the entire content of the manuscript, including any text generated or refined with the assistance of the LLM. We have ensured that all LLM-generated text adheres to ethical guidelines and does not contribute to plagiarism or scientific misconduct.

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

# A APPENDIX

## A.1 PROMPTS

**Iteration Prompt:** The prompt used in our iterative retrieval reasoning process for the phi-agent.

```
**User query**: {query}

**Graph evidence**: {context_text}

**TASK**: You are a reasoning assistant working with information
    from a knowledge graph.

Your task is to answer the user query with the provided graph
    evidence and your own knowledge.
Follow this process for each query:

1. Carefully read the user query and the retrieved graph evidence.
2. Think step-by-step about what the query is asking and what the
    evidence supports.
3. Use both the graph evidence and your own knowledge to reason
    through the query.

- If the evidence **is sufficient**, respond in the following format
    :
  <think> reasoning process here. </think>
  <label> able </label>
  <answer> final answer here. </answer>

- If the evidence **is not sufficient**, do the following:
  - Identify what key information is missing to answer the query.
  - Propose a natural, specific follow-up question that would help
    obtain the missing evidence.
  - Respond in the following format:
  <think> reasoning process here. </think>
  <label> unable </label>
  <query> proposed follow-up question here. </query>

Do not include any text outside the specified tags. Maintain strict
    adherence to the format.
```

**Force-answer Prompt:** The prompt that forces the model to answer when the preset maximum number of iterations is reached.

```
**User query**: {query}

**Graph evidence**: {context_text}

**TASK**: You are a reasoning assistant working with information
    from a knowledge graph.
```

```
Your task is to answer the user query with the provided graph
    evidence and your own knowledge.
Follow this process for each query:

1. Carefully read the user query and the retrieved graph evidence.
2. Think step-by-step about what the query is asking and what the
    evidence supports.
3. Use both the graph evidence and your own knowledge to reason
    through the query.

- then respond in the following format:
  <think> reasoning process here. </think>
  <answer> final answer here. </answer>

Do not include any text outside the specified tags. Maintain strict
    adherence to the format.
```

**Evelution Prompt:** The prompt used in LLM-based evelution.

```
Given the question and gold answer, please judge if the predicted
    answer is correct. Providing additional background or
    supplementary information is acceptable.

Question: {question}.

Gold Answer: {gold_answer}.

Predicted Answer: {answer}.

Please only return 1 (means correct) or 0 (means incorrect) in a
    concise way.
```

**Prompt in RL-data generation:** The prompt that generates RL data.

```
You are an evaluation assistant for knowledge-graph question
    answering.

Your task is to determine whether the gold answer can be logically
    inferred **only** from the question and the retrieved context.

Follow this reasoning protocol:

1. Carefully reason step by step about what the question requires
    and what information is present in the context.
2. Decide whether the context alone is sufficient to derive the gold
     answer.

If it is sufficient to infer the gold answer, output:
<think> ...your reasoning... </think>
<label> able </label>

If it is **not** sufficient, output:
<think> ...your reasoning... </think>
<label> unable </label>

---
```

```
User question:
{question}

Retrieved context:
{retrieved_context}

Gold answer:
{gold_answer}
```

```
Question: {question}
Retrieved Context: {retrieved_context}
Gold Answer: {gold_answer}
Supporting Facts: {supporting_facts}

---

You are a reasoning assistant for a knowledge-graph question
    answering system.

You are given the full information for a QA task, including:

- **Question**: The original question asked by the user, which the
    system is expected to answer;
- **Retrieved Context**: Relevant information retrieved from the
    knowledge graph, which may contain partial facts or background
    but is currently insufficient to answer the question;
- **Gold Answer**: The ground-truth answer labeled in the dataset;
- **Supporting Facts**: The original textual evidence that supports
    the Gold Answer. These help you understand the reasoning target,
     but you are **not allowed to directly quote or use them** in
    the generated question.

### Task:

The Gold Answer **cannot** be inferred from the Question and
    Retrieved Context alone.

Your job is to:

1. Carefully read the Question and Retrieved Context, and identify
    what **critical information is missing** in order to correctly
    answer the original question;
2. Propose a **natural and reasonable follow-up question** that
    would help acquire the missing information and thus support
    answering the original question.

### Constraints:
- The follow-up question must be something that can be **reasonably
    proposed based only on the Question and Retrieved Context**;
- You are **not allowed to directly copy from the Gold Answer or
    Supporting Facts**;
- Output should follow this format:

<think> Explain what key information is missing and how you
    determined this based on the Question and Retrieved Context. </
    think>
<query> Write your follow-up question here. </query>
```

