# OpenReview forum: "Phi-agent: A Dynamic In-Graph Reasoning Agent Inspired by the Prefrontal–hippocampal Interaction"
_ICLR.cc/2026/Conference — ICLR 2026 Conference Withdrawn Submission_

### Official Review · Reviewer_9tjQ · 2025-10-26

**Soundness:** 3
**Presentation:** 1
**Contribution:** 2
**Rating:** 2
**Confidence:** 4

**Summary:**

The paper introduces Phi-Agent, a reinforcement learning (RL) approach for multi-turn graph-based retrieval over text. Given a knowledge graph (KG) constructed from textual data, Phi-Agent employs trajectory-based RL with rewards based on correct final answer generation and the provision of helpful intermediate subqueries during the retrieval process. The method uses GRPO training to adapt Qwen3-1.7B for agentic retrieval tasks. Experimental results demonstrate Phi-Agent's effectiveness compared to non-agentic baselines such as HippoRAG.

**Strengths:**

- S1) Table 1 demonstrates that Phi-Agent achieves superior performance compared to non-agentic baselines, highlighting the benefits of RL and multi-turn retrieval across the evaluated benchmarks.

- S2) Ablation studies (Figure 4) reveal a key advantage of Phi-Agent: while standard agentic retrieval degrades with increased turns, Phi-Agent's RL technique mitigates this limitation, demonstrating greater robustness during extended interactions.

**Weaknesses:**

- W1) The paper lacks key novelties as it applies well-established RL techniques (Search-R1; Jin et al. 2025, Graph-R1; Luo et al. 2025) to HippoRAG's retrieval system. The authors should better justify why combining RL with HippoRAG is more beneficial than other RL approaches, such as non-graph-based methods like Search-R1. Additionally, the motivation in Section 3.1 appears similar to HippoRAG's hippocampus memory-inspired retrieval rationale.

- W2) The paper lacks comparison with stronger baselines including agentic retrieval methods (IRCOT; Trivedi et al., 2022) and RL-based techniques like Search-R1. Notably, Phi-Agent is evaluated primarily against single-turn retrieval systems, which does not provide sufficiently comprehensive evaluation.

- W3) It remains unclear how Phi-Agent integrates with other methods such as Dalk, LightRAG, GRetriever, and KGP in Table 2. The paper suggests Phi-Agent is specifically tailored for HippoRAG (lines 158-159), so the authors should provide more detailed experimental setup information.

**Questions:**

- Q1) What is the Response Token Cost for single-turn baselines, such as HippoRAG in Table 3?

- Suggestion: The paper would benefit from further polishing, including fixing typos, adding appropriate citations for experimental baselines, and correcting citation formatting.

---

### Official Review · Reviewer_vPfc · 2025-10-30

**Soundness:** 3
**Presentation:** 2
**Contribution:** 3
**Rating:** 4
**Confidence:** 4

**Summary:**

The paper proposes Phi-agent, a brain-inspired dynamic graph reasoning agent modeled after the prefrontal–hippocampal interaction mechanism. It performs iterative “retrieve–reason–re-retrieve” cycles to enable self-reflective, multi-round in-graph retrieval and reasoning, overcoming the limitations of static, one-shot GraphRAG approaches. The authors design a joint reinforcement learning reward mechanism that integrates both reasoning quality and retrieval trajectory, and train a Qwen3-1.7B model on a dataset of 7,405 samples. Experiments on HotpotQA, MuSiQue, and 2Wiki show that Phi-agent significantly outperforms existing methods in both accuracy and efficiency, highlighting a promising neuro-inspired direction for graph-based reasoning systems.

**Strengths:**

1. This paper tackles a key limitation of existing GraphRAG methods—their use of single-round, static retrieval that underutilizes the depth of knowledge graphs. The authors propose a multi-round retrieval framework where an agent dynamically decides whether further retrieval is needed after each reasoning step.

2. The agent is trained using synthetic data and reinforcement learning with a joint reward mechanism that balances reasoning accuracy and retrieval trajectory optimization.

3. Experiments on HotpotQA, MuSiQue, and 2WikiQA show that the proposed method achieves consistent performance gains across different GraphRAG variants and backbone models, demonstrating strong generalization and practical value.

**Weaknesses:**

While the paper presents a well-executed system and strong empirical results, there are several concerns regarding **methodological novelty** and **experimental consistency** that should be addressed before publication.

---

#### **1. Limited Methodological Novelty**

The methodological innovation of the paper appears **relatively limited**.

* The overall framework builds heavily on **HippoRAG** (Jimenez Gutierrez et al., 2024), inheriting its graph construction and one-shot retrieval mechanisms.
* The proposed multi-round retrieval strategy employs LLMs to determine whether additional retrieval is needed. However, this stragety has been proposed in some **earlier iterative RAG and LLM-based KGQA** frameworks.
* The reinforcement learning setup based on **Group Relative Policy Optimization (GRPO)** (Shao et al., 2024) is a fairly standard choice for policy optimization in RAG-style agents.

Therefore, the paper can be viewed as a **well-integrated combination** of existing techniques—graph-based retrieval (HippoRAG), multi-turn retrieval control, and GRPO training—rather than introducing a fundamentally new theoretical contribution. The strong experimental performance demonstrates good engineering and system design but does not yet establish a **clear theoretical advancement** in graph-based retrieval-augmented reasoning.

---

#### **2. Experimental Inconsistencies and Ambiguities**

Several issues in the experimental section require clarification:

* In **Table 1**, the reported *top-20 accuracy* for the Wiki dataset (23.6%) under “Ours” seems **unusually low**, especially compared to other datasets, and may indicate a data or evaluation mismatch.
* **Metric inconsistency**: Table 1 reports *top-10* and *top-20* accuracies, while Table 2 uses *top-5* accuracy. The reason for this discrepancy is not explained, and it makes cross-table comparison difficult.
* The reported *top-5* accuracy for **HippoRAG** being slightly higher than its *top-10* accuracy is **counterintuitive**, suggesting a possible evaluation or reporting issue.
* The paper lacks **comparative ACC results** obtained with different Phi-agent backbones—e.g.,  untrained **Qwen3-1.7B**, or other LLMs (e.g., Qwen-4B, GPT-4o) under the same conditions—to assess the generality and stability of the proposed training approach across models.

Overall, while the results are promising, these inconsistencies reduce confidence in the experimental rigor and reproducibility. A more transparent and standardized evaluation protocol would strengthen the paper significantly.

---
#### **3. Typos**
Line 373: without any predefined iteration limits **nder** the retrieval

---

#### **References**

* Jimenez Gutierrez, B. et al. (2024). *HippoRAG: Neurobiologically inspired long-term memory for large language models.* NeurIPS 2024.
* Shao, Z. et al. (2024). *DeepSeekMath: Pushing the limits of mathematical reasoning in open language models.* arXiv:2402.03300.

**Questions:**

Please see the weaknesses part.

---

### Official Review · Reviewer_9ae1 · 2025-10-31

**Soundness:** 3
**Presentation:** 3
**Contribution:** 1
**Rating:** 2
**Confidence:** 4

**Summary:**

The paper introduces Phi-agent, which utilizes a "retrieve-reason-reretrieve-rereason" loop framework. The agent is post-trained with GRPO, using rewards including trajectory, answer accuracy, and formatting correctness. A dataset is constructed for the training, and experiments show that Phi-agent can improve downstream QA acc compared to baselines.

**Strengths:**

- The idea is straightforward, and the method design makes sense.
- The dataset constructed can be valuable for other works.
- Experiments show good improvements.

**Weaknesses:**

- Though starting with a fancy story connected to prefrontal-hippocampal Interaction, the key framework seems like a subset of ReAct.
- The training framework, trajectory + answer accuracy + formatting correctness + GRPO, is somewhat with limited technical novelty.
- The experiment results seem to be mostly from LLMs. Is EM/F1 reported as previous works?
- How can the trained agent generalize across datasets?

**Questions:**

See above.

---

### Official Review · Reviewer_PP7J · 2025-11-01

**Soundness:** 3
**Presentation:** 3
**Contribution:** 2
**Rating:** 4
**Confidence:** 5

**Summary:**

This paper proposes Phi-agent, a brain-inspired dynamic in-graph reasoning agent for GraphRAG, addressing static one-shot retrieval limitations via a "retrieve–reason–re-retrieve" loop (PPR simulates hippocampal memory activation, language model mimics prefrontal reasoning). It introduces a joint reward mechanism, constructs a 7,405-sample dataset, fine-tunes Qwen3-1.7B with GRPO, and achieves SOTA on HotpotQA, MuSiQue, and 2WikiQA—small models even outperform larger baselines. Key contributions: Phi-agent design, joint reward/dataset, and experimental validation.

**Strengths:**

1. Originality: Integrates prefrontal-hippocampal interaction into GraphRAG, enabling dynamic context construction via iterative loops—an innovative neurobiology-RAG combination.
2. Quality: Rigorous dataset annotation (multi-model voting) and unified experimental setup (consistent Qwen backbones, same embedding model) ensure reliable, fair baseline comparisons.
3. Clarity: Clear descriptions of Phi-agent workflow, reward design, and RL training; figures/tables effectively illustrate methods and results.
Significance: Boosts multi-hop reasoning performance; small models (e.g., 4B HippoRAG-ϕ) outperform large ones (32B HippoRAG), reducing deployment costs.

**Weaknesses:**

1. Insufficient mapping between PPR and real hippocampal mechanisms—only functional analogy, no clarity on core mechanisms (e.g., association strength calculation alignment).
2. Joint reward function’s α/β/γ determination (e.g., grid search, empiricism) is unspecified, lacking transparency.
3. No analysis of PPR efficiency for large-scale knowledge graphs (e.g., 1M entities) or optimization strategies (subgraph pruning, indexing).
4. Incomplete sub-query evaluation—only semantic similarity to gold queries is measured, ignoring actual retrieval value (e.g., failing to fetch key evidence).
5. No discussion of knowledge graph quality impact (e.g., uneven entity density) on PPR retrieval and Phi-agent performance.
6. Missing comparisons with similar complex correlation modeling RAG methods (e.g., Hyper-RAG, HiRAG).

**Questions:**

see weaknesses

---

### Note · Authors · 2025-12-09

I have read and agree with the venue's withdrawal policy on behalf of myself and my co-authors.